# Conditional and interaction gene-set analysis reveals novel functional pathways for blood pressure

Christiaan A. de Leeuw [1], Sven Stringer [1], Ilona A. Dekkers [2], Tom Heskes[3] & Danielle Posthuma[1,4]

Gene-set analysis provides insight into which functional and biological properties of genes are aetiologically relevant for a particular phenotype. But genes have multiple properties, and these properties are often correlated across genes. This can cause confounding in a gene-set analysis, because one property may be statistically associated even if biologically irrelevant to the phenotype, by being correlated with gene properties that are relevant. To address this issue we present a novel conditional and interaction gene-set analysis approach, which attains considerable functional refinement of its conclusions compared to traditional gene-set analysis. We applied our approach to blood pressure phenotypes in the UK Biobank data ($N = 360,243$), the results of which we report here. We confirm and further refine several associations with multiple processes involved in heart and blood vessel formation but also identify novel interactions, among others with cardiovascular tissues involved in regulatory pathways of blood pressure homoeostasis.

[1] Department of Complex Trait Genetics, Center for Neurogenomics and Cognitive Research, Amsterdam Neuroscience, VU University Amsterdam, Amsterdam 1081 HV, The Netherlands. [2] Department of Radiology, Leiden University Medical Center, Leiden 2333 ZA, The Netherlands. [3] Institute for Computing and Information Sciences, Radboud University Nijmegen, Nijmegen 6525 EC, The Netherlands. [4] Department of Clinical Genetics, Amsterdam Neuroscience, VU University Medical Center, Amsterdam 1007 MB, The Netherlands. Correspondence and requests for materials should be addressed to C.Leeuw. (email: c.a.de.leeuw@vu.nl) or to D.P. (email: d.posthuma@vu.nl)

The aim of gene-set analysis (GSA) is to uncover functional and biological properties of genes involved in the genetic aetiology of a phenotype[1,2]. If a property is relevant to a phenotype, then variants associated with that phenotype will tend to accumulate in genes with that property. For example, smooth muscle cells (SMC) play a role in blood pressure regulation[3,4], and if this has a genetic basis we might expect to find genes involved in the development of SMCs to exhibit genetic association with blood pressure phenotypes.

However, genes typically have numerous different properties, which can be strongly correlated with each other if they involve many of the same genes. Perhaps SMC development genes are also involved in the development of other types of muscle cell, or they are expressed primarily in muscle tissue. This would result in a correlation between SMC development and muscle cell development in general, or between SMC development and muscle-specific gene expression.

In such scenarios, associated variants will accumulate in genes with a property that does not itself play a role in the phenotype, but is correlated with another gene property that does. Thus, the SMC development gene set could become associated simply by muscle-specific gene expression playing a role in the phenotype. Traditional GSA only tests the marginal associations of gene properties[5,6], and cannot account for this kind of confounding. Such GSA is therefore liable to identify gene properties that hold no biological relevance for the phenotype, with potentially very misleading interpretations and wasted effort in follow-up research as a result.

To address this issue we have developed a novel GSA approach, based on and implemented in our existing GSA tool MAGMA[5]. Central to this approach is the conditional GSA model, which can evaluate how associations of different gene properties relate to each other. As Fig. 1 illustrates, it can identify confounding where traditional GSA cannot.

The model can also deal with more complex scenarios, in which particular combinations of multiple gene properties are relevant to the phenotype rather than any individual gene properties on their own. This manifests statistically as an interaction between gene properties, which are hard to detect when testing only marginal associations and which can result in confounding of the gene properties involved (Fig. 1d). A more complete and accurate insight into the phenotype based on GSA therefore requires that such scenarios are taken into account as well.

Our proposed approach works by selecting gene properties with significant marginal associations using a standard GSA, then using a series of follow-up analyses to discard those which are likely not biologically relevant for the phenotype. A wide range of gene properties is used as input to improve the probability of relevant gene properties being included, as this allows for the detection of confounding caused by those relevant gene properties. This also improves the specificity of the conclusions that can be drawn because more gene properties can be ruled out as having no biological relevance to the phenotype, and an absence of confounding where it might have been expected can also be shown.

The analysis workflow for our approach is shown in Fig. 2, with a more detailed overview of this analysis workflow provided in the Methods section and a guide to performing and interpreting the analysis in the Supplementary Methods. The initial GSA in step 1 can include both binary sets and continuous gene-level variables, and is followed by four follow-up analysis steps that refine the initial results. The results are first corrected for global effects that are likely to act as general confounders in the GSA (e.g., gene expression levels), after which overlap between significant associations is evaluated. For gene sets (i.e. binary gene properties) this is followed by additional checks for outliers and signs of further confounding. Finally, post hoc interaction analyses are performed for all significant gene properties, to refine the interpretation of their effects. In an optional sixth step, exploratory interaction analysis is applied to detect additional associations that were not picked up in the initial GSA.

We performed a simulation study to validate the conditional and interaction GSA models used in our workflow, and then to demonstrate the analysis workflow we applied it to the analysis of blood pressure phenotypes. For this we used the UK Biobank[7] data, analysing three blood pressure phenotypes: systolic blood pressure (SBP), diastolic blood pressure (DBP) and pulse pressure (PP). The gene annotation used in these analyses consisted of gene sets from the three Gene Ontology domains[3,8], miRNA target gene sets[9], and continuously valued tissue-specific gene expression levels from the GTEx data[10]. A replication study was also performed to further validate results from the UK Biobank analysis.

High blood pressure is an important risk factor for cardiovascular disease[11] and has an estimated heritability of 30–50%[12]. Recently, large-scale GWAS studies have identified over 400 loci that regulate blood pressure[10,13–17], with many of the identified loci showing associations with different blood pressure phenotypes[16]. Some GSA was performed as part of these studies, but only to a limited extent (see Supplementary Methods for a brief overview) and only using traditional GSA approaches. Applying our extend GSA analysis workflow to these phenotypes may therefore expand our current understanding of the genetic aetiology and biological mechanisms of blood pressure regulation.

Our analyses show that confounding and overlap between associations is widespread, with the majority of initially significant associations found to be due to the effects of general confounders and the associations of other gene properties. Interactions are also prevalent and often involve gene properties with no detectable marginal association, suggesting that the interaction analysis model can provide additional insights into the phenotype to complement those of standard GSA. For the blood pressure phenotypes a range of processes involved in heart and blood vessel formation have been identified, as well as tissue-specific expression in artery, heart and female reproductive organs. Several novel interactions have also been found, among others identifying joint involvement of cardiovascular development and homoeostatic processes, and involvement of heart-expressed miRNA-145 target genes.

## Results

**Simulations demonstrate risk of confounding in GSA.** A simulation study was performed to evaluate the conditional and interaction GSA models, both individually and in relation to the standard marginal GSA (details on the simulation settings are provided in the Methods and Supplementary Methods). As shown in Supplementary Figure 1, marginal GSA is highly vulnerable to confounding. When a gene set with no biological effect assigned to it is analysed, it will be statistically significant at a rate far exceeding the significance threshold if it overlaps with another gene set that does have an effect. The conditional analysis model can effectively account for this however, correcting for the confounding effect of the overlapping set and yielding an error rate at the nominal significance level. This phenomenon is also clearly illustrated in the blood pressure analyses, for example for the heart development gene set. For PP it is initially significant, with a marginal $p$-value of $1.6 \times 10^{-6}$. This association is entirely explained by the much stronger association of the cardiovascular system development set that contains it, with a conditional $p$-value for heart development of only 0.40.

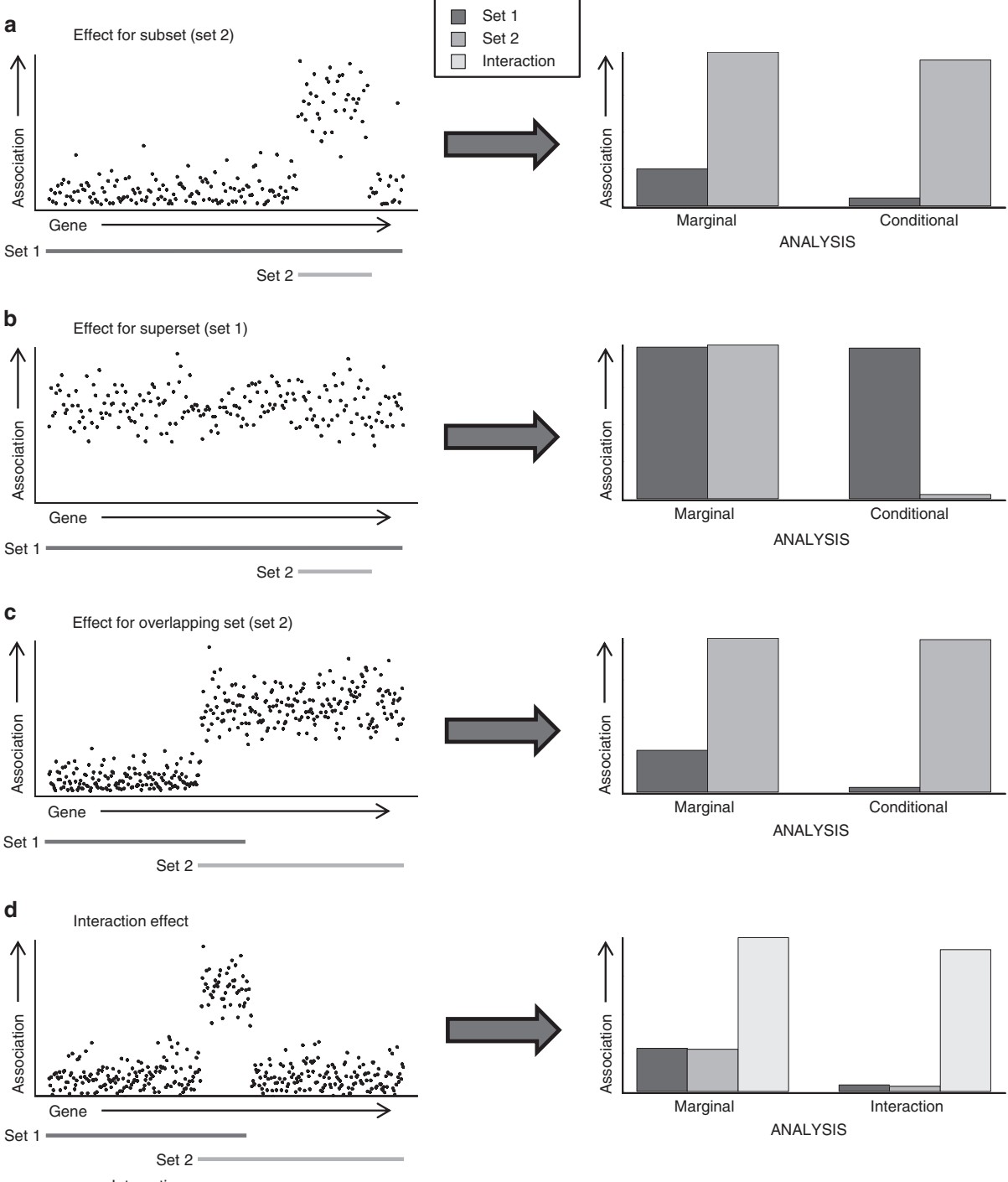

**Fig. 1** Illustration of different confounding scenarios. In each of the scenarios, a gene set with no relevance to the phenotype overlaps with a relevant gene set, resulting in a confounded association. The left column contains scatterplots of gene associations as a function of their position, with the lines below the plots indicating which gene sets they belong to. The right column contains bar plots showing the resulting gene-set associations, when analysing either the marginal associations of each gene set individually (as in traditional GSA) or when using a joint conditional (**a–c**) or interaction (**d**) analysis of the two gene sets. In **a–c**, one of the gene sets is relevant to the phenotype, with the other having no effect; in **d** the effect is assigned to the interaction between the two sets, with neither having a main effect. In all scenarios this is shown to be correctly reflected by the conditional/interaction analysis, but not the marginal analyses

A similar situation is shown in Supplementary Figure 2. Here, two overlapping gene sets were simulated, with one or both of them assigned an effect. This was then analysed in two ways: analysing the two gene sets and their interaction in an interaction GSA, and analysing the interaction set (containing all genes shared by the two gene sets) by itself with a marginal GSA. In these simulations there are no actual interaction associations, and for the interaction analysis the error rates are indeed at the

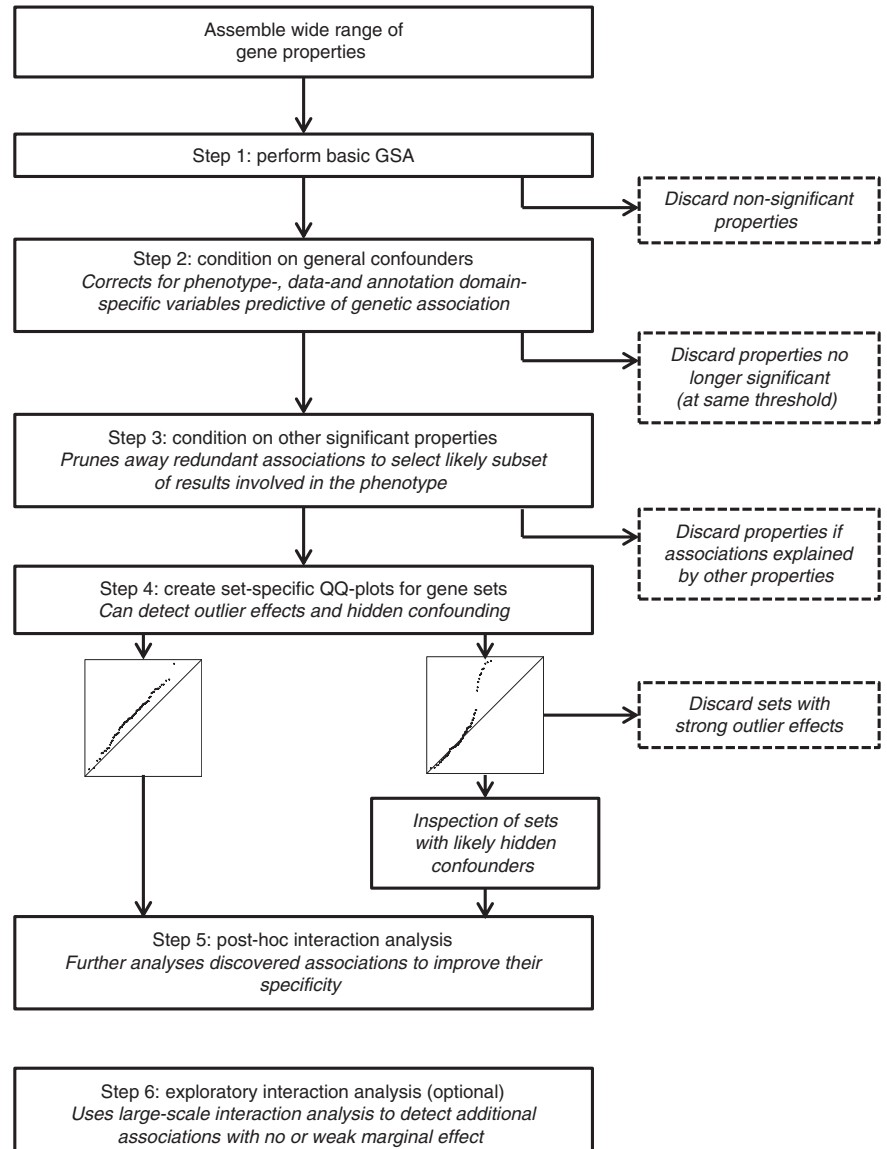

**Fig. 2** Overview of the extended gene-set analysis workflow. The workflow is composed of five steps, starting with a standard gene-set analysis in step one. Results from this analysis are then successively refined in the subsequent steps, discarding initially significant gene properties if their associations are found to be insufficiently robust. In the optional sixth step an exploratory interaction analysis is used to detect additional interaction effects not uncovered in the main analysis workflow

nominal significance level. When testing the marginal association of the interaction set however, the error rates are strongly inflated.

Although normally an interaction would not be analysed in this way, it can happen that gene sets are defined in terms of a combination of multiple gene properties. For example, a gene set may be defined as all the genes in a particular pathway that are also differentially expressed in the heart. Such a gene set is therefore actually an interaction between that pathway and differential heart expression, and will be confounded by any main effects that the pathway or differential heart expression may have. For these kinds of compound gene set, the interaction GSA is therefore required as well.

**Gene–property associations are strongly overlapping.** Results of the blood pressure analyses at different steps of the workflow are summarised in Table 1, with the individual associations retained at the end of the workflow shown in Table 2. Initially significant

associations that were later discarded can be found in Supplementary Tables 1 and 2. As shown there is a considerable reduction in the number of associations in the final results, compared to the standard GSA in step 1.

A large portion of this is due to the general confounders that are corrected for in step 2, which reduced the number of associations by 75%. Conditioning the remaining associations on each other in step 3 led to a further reduction of 30%. Moreover, there were multiple instances of gene properties being selected jointly, with their associations clearly reflecting a single signal but their overlap too strong to be disentangled. The number of distinct signals captured by these significant and retained gene properties is therefore even lower.

This suggests the presence of a great deal of overlap between the associations in the standard GSA, with many of the tested gene properties tapping into a much smaller subset of shared signals. Moreover, in practice the overlap in associations among

**Table 1 Overview of the number of initially significant gene properties retained in each step**

| Domain | Number of gene properties | Number of significant and retained gene properties | | | | |
|---|---|---|---|---|---|---|
| | | Phenotype | Step 1 | Step 2 | Step 3[a] | Final[a] |
| Tissue-specific gene expression | 53 | SBP | 52 | 1 | 1 | 1 |
| | | DBP | 50 | 3 | 3 (2) | 3 (2) |
| | | PP | 43 | 14 | 7 (3) | 7 (3) |
| miRNA targets | 221 | SBP | 0 | 0 | 0 | 0 |
| | | DBP | 0 | 0 | 0 | 0 |
| | | PP | 3 | 0 | 0 | 0 |
| GO—biological process | 4653 | SBP | 10 | 6 | 5 | 5 |
| | | DBP | 14 | 6 | 5 (4) | 5 (4) |
| | | PP | 31 | 16 | 9 (8) | 9 (8) |
| GO—cellular component | 584 | SBP | 1 | 1 | 1 | 1 |
| | | DBP | 0 | 0 | 0 | 0 |
| | | PP | 6 | 2 | 2 (1) | 2 (1) |
| GO—molecular function | 929 | SBP | 2 | 1 | 1 | 0 |
| | | DBP | 1 | 1 | 1 | 1 |
| | | PP | 6 | 2 | 2 | 2 |
| All domains | 6440 | SBP | 65 | 9 | 8 | 7 |
| | | DBP | 65 | 10 | 9 (7) | 9 (7) |
| | | PP | 89 | 34 | 20 (14) | 20 (14) |
| | | Combined | 219 | 53 | 37 (29) | 36 (28) |

Gene properties are initially included based on significance in step 1, then retained or discarded in subsequent steps
Multiple testing correction was performed separately for each phenotype, applying Bonferroni correction per domain with $\alpha = 0.05/5 = 0.01$
SBP systolic blood pressure, DBP diastolic blood pressure, PP pulse pressure, GO Gene Ontology
[a] Numbers in parentheses reflect the likely number of distinct underlying signals

different gene properties in particular is even stronger than the reduction in the number of hits suggests, as shown in Fig. 3a. Looking at the associations of all the gene sets, the effect of conditioning on general confounders is relatively moderate and primarily affects the strongest associations. However, conditioning on all the significant gene sets retained at the end of the analysis workflow has a much more profound impact. It is most pronounced for PP, for which almost no marginal association remains, but it strongly affects the other two phenotypes as well (Supplementary Figure 3).

**Evidence of widespread gene-set interaction**. The analyses show that although not as extensive as for the marginal associations, there is considerable evidence for interactions both between pairs of gene sets (Fig. 3b, Supplementary Figure 4) and between gene sets and tissue-specific gene expression (Fig. 3c, Supplementary Figure 5). This is also reflected in the individual results for the post hoc interaction analyses, with significant interactions of both kinds (Tables 3 and 4). It seems unlikely that this is unique to blood pressure phenotypes, which suggests that gene properties probably commonly affect the phenotypes specifically in combination with other gene properties. It follows that finding these interactions is necessary for gaining a proper insight into the genetics of a phenotype.

In the post hoc analyses, by definition one of the gene properties had a marginal association strong enough to be detected. In some cases this may reflect a genuine main effect, but this can also happen when there is only a strong interaction. An example of this is the cell proliferation gene set, for which the marginal association can be entirely explained by two interactions (see below). For the majority of interactions found in the post hoc analyses, the second gene property also shows little or no evidence of any marginal association. The involvement of those gene properties would therefore be very difficult to detect in a normal GSA. The exploratory interaction results point to the same conclusion, with for many of the gene properties involved in

the top interactions again little evidence of marginal associations (Supplementary Table 4).

It is also clear that such weak marginal associations can hide very strong effects. For the interactions between tissue expression and gene sets, the p-values of the subset of top 25% expressed genes are often very low. Similarly, for the top interactions found in the exploratory analysis, three of the four negative interactions hide significant main effects of gene properties that are not marginally significant. Although for these the observed marginal associations were stronger, they were still not strong enough for the GSA in step 1 to detect them. Since negative interactions are relatively prevalent (Fig. 3c), this again suggests that there may be a considerable amount of association that a normal GSA cannot easily uncover.

**Variability across gene-set domains**. In our results there are considerable differences between the Gene Ontology and miRNA target gene-set domains, in both the number of significant results (Table 1) and the overall levels of association (Supplementary Figure 6). The majority of the significant results are found in the Gene Ontology biological process domain, with only a handful of additional associations in the cellular component and molecular function domains. For the miRNA target sets, no associations are found at all.

As Supplementary Figure 6 shows, the miRNA results are not entirely devoid of signal, and the general class of miRNA target genes shows a strong association for both SBP and PP (Supplementary Table 5). No individual miRNA families emerge from the analysis however, with the three initially significant miRNA target set associations explained away by the gene expression and general miRNA target gene effects. It may be that the miRNA target sets are too broad and are not involved in the phenotypes as a whole, a possibility supported by the strong interaction found for miRNA-145 with heart expression (see below).

Although the cellular component and molecular function domains do yield some associations they are dominated by the

**Table 2 Marginally significant gene properties retained at end of extended analysis workflow**

| Gene property | No. of genes | Phenotype | p-value | Shared | QQ check |
|---|---|---|---|---|---|
| **Tissue-specific gene expression** | | | | | |
| Artery (aorta) | – | PP | 1.16e-11 | (1) | – |
| Artery (coronary) | – | DBP | 0.000113 | (2) | – |
| Artery (coronary) | – | PP | 3.13e-10 | (1) | – |
| Artery (tibial) | – | DBP | 8.82e-5 | (2) | – |
| Artery (tibial) | – | PP | 1.15e-11 | (1) | – |
| Cervix (endocervix) | – | PP | 1.58e-6 | (3) | – |
| Heart (atrial appendage) | – | PP | 2.26e-8 | | – |
| Ovary | – | PP | 5.65e-6 | (3) | – |
| Uterus | – | SBP | 0.000123 | | – |
| Uterus | – | DBP | 4.78e-5 | | – |
| Uterus | – | PP | 4.03e-8 | (3) | – |
| **GO—biological process** | | | | | |
| Blood vessel remodelling | 29 | DBP | 5.74e-7 | | |
| Cardiocyte differentiation | 95 | SBP | 6.28e-9 | | |
| Cardiocyte differentiation | 95 | PP | 9.49e-9 | | |
| Cardiovascular system development | 764 | PP | 1.79e-9 | (4) | |
| Cell proliferation | 645 | PP | 1.60e-8 | | |
| CGMP biosynthetic process | 13 | DBP | 4.06e-7 | | |
| Circulatory system development | 764 | PP | 1.79e-9 | (4) | |
| Embryonic eye morphogenesis | 32 | SBP | 4.17e-7 | | |
| Mesenchyme development | 180 | PP | 9.11e-9 | | |
| Negative regulation of cellular senescence | 36 | PP | 3.63e-8 | | |
| Negative regulation of smooth muscle cell proliferation | 11 | PP | 6.20e-7 | (5) | |
| Negative regulation of transcription from RNA polymerase II promotor | 701 | SBP | 9.22e-7 | | Flagged |
| Nitric oxide metabolic process | 15 | DBP | 6.22e-8 | (6) | |
| Positive regulation of developmental growth | 150 | SBP | 1.84e-6 | | |
| Positive regulation of urine volume | 14 | SBP | 6.43e-7 | | |
| Positive regulation of urine volume | 14 | DBP | 5.72e-7 | | |
| Reactive oxygen species biosynthetic process | 21 | DBP | 3.37e-8 | (6) | |
| Regulation of smooth muscle cell proliferation | 98 | PP | 1.00e-7 | (5) | |
| Regulation of transcription from RNA polymerase II promotor | 1682 | PP | 9.97e-7 | | |
| **GO—cellular component** | | | | | |
| Actin cytoskeleton | 430 | PP | 1.08e-6 | (7) | Flagged |
| Cytoskeleton | 1882 | PP | 1.88e-7 | (7) | Flagged |
| T tubule | 45 | SBP | 1.35e-5 | | Flagged |
| **GO—molecular function** | | | | | |
| Cell adhesion molecule binding | 180 | PP | 1.28e-6 | | Flagged |
| Peptide hormone binding | 35 | PP | 4.96e-6 | | Flagged |
| Sequence-specific DNA binding | 976 | DBP | 5.62e-6 | | |

p-Values are from step 2 of the analysis workflow, after correcting for general confounders. Gene properties that likely reflect a single shared association are marked by the same number in the 'Shared' column. Gene sets for which issues were noted during inspection of the QQ-plots are marked in the 'QQ check' column. These are still valid, but require more caution when interpreting their association
SBP systolic blood pressure, DBP diastolic blood pressure, PP pulse pressure, GO Gene Ontology

biological process domain, a feature that is found in the interaction analyses as well (see Tables 3 and 4, Supplementary Table 4). This is in part due to there being significantly more biological process gene sets to analyse, though for the marginal associations this is compensated by the correspondingly more stringent multiple testing correction. Moreover, the associations that are found for cellular component and molecular function are not entirely convincing, with almost all of them showing irregularities in their set-specific QQ-plots (Table 2, Supplementary Figures 7 and 8; see also step 4 of the detailed analysis overview in the Supplementary Methods).

**Tissue expression predicts blood pressure association.** Initial analysis of the GTEx gene expression levels shows that overall gene expression is significant for all three phenotypes (Supplementary Table 5), meaning that genes with a higher average gene expression tend to have stronger genetic associations with the phenotypes as well. This general effect drives the associations found for many of the individual tissues, with the majority of the

associations for these tissues disappearing once the general effect is corrected for (Table 1).

Expression in the remaining tissues is still strongly correlated however, making it difficult to attribute associations to any individual tissue. Conditioning the tissues on each other suggests that there are likely at most three distinct clusters of association (see Table 2). The first and strongest is in the arterial expression levels, present in both DBP and PP. This arterial association explains a large proportion of the other tissue associations, but a second cluster of female reproductive organs remains. It is the only association common to all three phenotypes, and manifests most prominently in the uterus expression. Unique to PP there is also a third association, however, for heart (atrial appendage) expression.

**Tissue-expression dependency of gene-set associations.** Post-hoc interaction analyses for the tissue-specific expression shows that there is also considerable positive interaction between tissue expression levels and gene sets, with significant interactions for all

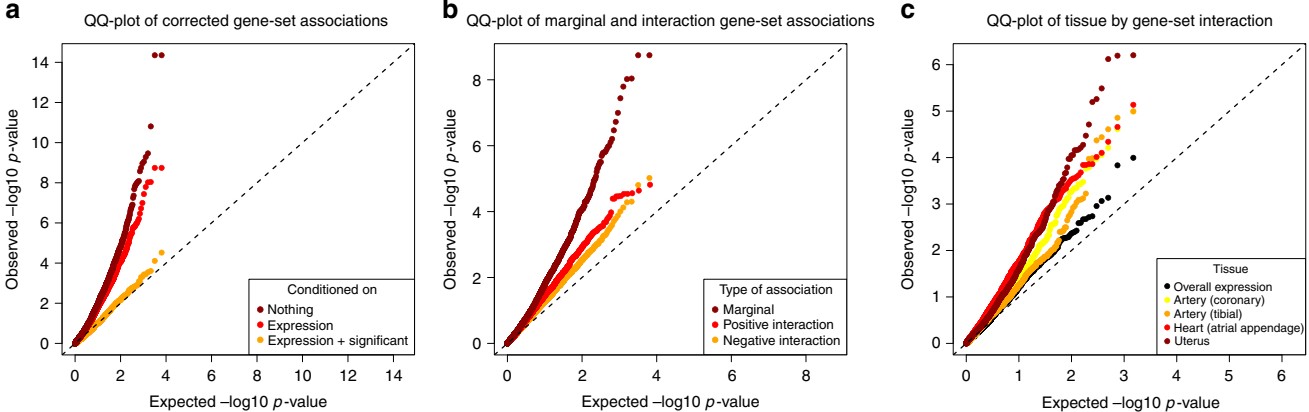

**Fig. 3** Global QQ-plots of associations for pulse pressure. **a** Comparison of gene-set associations corrected for other effects. Shown are associations with no corrections (step 1), associations corrected for overall and tissue-specific gene expression (step 2), and associations additionally corrected for all significant and retained gene sets listed in Table 2. When correcting for gene expression, the associations for miRNA target sets were also corrected for general miRNA target status. **b** Comparison of the overall levels of marginal and interaction association. Marginal associations are corrected for general confounders (step 2); the interaction associations are from the exploratory interaction analysis (step 6). **c** Comparison of tissue by gene set interactions for different tissues. For all of the tissue-specific analyses, each interaction was also conditioned on the interaction between overall expression and the gene set. For all three plots, corresponding figures for all of the phenotypes can be found in Supplementary Figures 3–5

of the four analysed tissues and all three phenotypes (Table 3, Fig. 4). The level of interaction association is found to vary across phenotypes and tissues and also seems to be tissue-specific, as there is little sign of interaction for the overall gene expression level (Fig. 3c).

Positive interaction between tissue-specific expression and a gene set represents a scenario where there is an association specific to the more strongly expressed genes in the gene set. Many of the gene sets involved are quite different in function from those found in the main GSA, and have generally weak marginal associations. One finding is a set of interactions between uterus-specific expression and three biological processes relating to sexual development for both SBP and PP, most strongly found for sex differentiation (interaction $p$-values of $5.2 \times 10^{-6}$ and $6.3 \times 10^{-7}$, respectively). Marginal associations for sex differentiation have $p$-values of only 0.0034 and 0.0052, respectively, but when the subset of more strongly uterus-expressed genes is tested, strong associations emerge (conditional $p$-values of $2.2 \times 10^{-6}$ and $4.3 \times 10^{-7}$). This effect is specific to uterus expression, with no sign of interaction for the other analysed tissues.

Another novel finding is the interaction between tibial artery expression and miRNA-145 target genes for SBP and PP (interaction $p$-values of $1.6 \times 10^{-5}$ and $1.0 \times 10^{-5}$). Marginal association is now absent altogether ($p$-values of 0.442 and 0.638), but again the subset of top expressed genes is highly significant (conditional $p$-values of $1.7 \times 10^{-7}$ and $6.2 \times 10^{-7}$). There are also several interactions between nucleotide, nucleoside and purine processes, and both arterial and heart expression, found for all three phenotypes.

One other surprising result is the interaction between heart expression regulation of blood pressure, highly significant for both SBP and DBP (interaction $p$-values of $2.6 \times 10^{-8}$ and $2.8 \times 10^{-9}$). It is also initially significant for PP, but the association is not as strong and does not survive the outlier correction (Supplementary Table 3). What makes this result surprising is that, in an analysis of blood pressure phenotypes, it only shows up here. It has no marginal associations, nor do any of its subsets, and also does not interact with artery expression. Yet in conjunction with heart expression its associations are very strong, with conditional $p$-values for the top expressed subset ($1.6 \times 10^{-9}$

and $2.0 \times 10^{-11}$ respectively) lower than for any of the marginal gene-set associations.

**Cardiovascular and muscle cell involvement**. For PP, a number of different biological processes related to the heart were found to be associated (Table 2). The strongest of these was cardiovascular system development ($p = 1.8 \times 10^{-9}$) (or circulatory system development, which is identical), which by itself accounts for much of the association of the other heart-related processes.

The association for cardiovascular system development is in turn partly explained by its two significant interactions (see Table 4), with the nested sets chemical homoeostasis and homoeostatic process (interaction $p$-values of $1.4 \times 10^{-5}$ and $2.0 \times 10^{-5}$). These interactions explain part of the marginal cardiovascular system development association (main effect $p$-values of 0.00067 and 0.00098 in the interaction model), but enough of it remains to suggest that its joint effect with the homoeostasis gene sets is important but is not the whole story of its role in blood pressure genetics.

There is also evidence for a related involvement of muscle cell processes, with cardiocyte differentiation significant for both SBP and PP ($p$-values of $6.3 \times 10^{-9}$ and $9.5 \times 10^{-9}$) and another association for the nested pair of sets negative regulation of smooth muscle cell proliferation and regulation of smooth muscle cell proliferation for PP ($p$-values of $6.2 \times 10^{-7}$ and $1.0 \times 10^{-7}$).

**Role of cell proliferation and intracellular regulation**. Another strong association is found for the cell proliferation set for PP ($p = 1.6 \times 10^{-8}$). Although this set overlaps with the two SMC proliferation sets, it is much larger and represents an independent additional signal. This signal can be traced to a pair of two largely independent interactions of cell proliferation (see Table 4), with the biological processes regulation of intracellular transport and regulation of intracellular signal transduction (interaction $p$-values of $7.6 \times 10^{-6}$ and 0.00020). Although similar in their function, these two interactions do not strongly overlap. Jointly they do account for almost all of the marginal association of cell proliferation, with its main effect $p$-value reduced to 0.015 when conditioning on both interactions simultaneously.

**Table 3 Significant and retained interactions from post hoc tissue expression by gene set interaction analysis**

| Tissue | Gene set | Marginal (set) | Interaction | Top 25% | Shared |
|---|---|---|---|---|---|
| | | *p*-Value | | | |
| **SBP** | | | | | |
| Artery (coronary) | Nucleoside phosphate biosynthetic process (BP) | 0.0167 | 2.59e-8 | 3.86e-5 | (1) |
| Artery (coronary) | Purine-containing compound biosynthetic process (BP) | 0.0104 | 1.81e-7 | 6.38e-5 | (1) |
| Artery (tibial) | miRNA-145 targets | 0.442 | 1.60e-5 | 1.74e-7 | |
| Artery (tibial) | Nucleoside phosphate biosynthetic process (BP) | 0.0167 | 8.56e-7 | 2.28e-4 | (1) |
| Artery (tibial) | Purine-containing compound biosynthetic process (BP) | 0.0104 | 1.12e-6 | 1.79e-5 | (1) |
| Heart (atrial appendage) | Positive regulation of catalytic activity (BP) | 0.122 | 9.24e-7 | 8.53e-3 | (2) |
| Heart (atrial appendage) | Positive regulation of molecular function (BP) | 0.0238 | 3.25e-6 | 1.86e-3 | (2) |
| Heart (atrial appendage) | Receptor signalling protein activity (MF) | 0.123 | 2.39e-5 | 2.75e-5 | |
| Heart (atrial appendage) | Regulation of blood pressure (BP) | 0.0115 | 2.61e-8 | 1.57e-9 | |
| Heart (atrial appendage) | Vascular process in circulatory system (BP) | 0.00139 | 5.88e-9 | 2.25e-7 | |
| Uterus | Development of primary sexual characteristics (BP) | 0.00698 | 1.3e-5 | 3.19e-8 | (3) |
| Uterus | Reproductive system development (BP) | 0.000829 | 2.93e-5 | 4.71e-4 | (3) |
| Uterus | Sex differentiation (BP) | 0.00339 | 5.21e-6 | 2.21e-6 | (3) |
| **DBP** | | | | | |
| Artery (coronary) | Nucleoside phosphate biosynthetic process (BP) | 0.550 | 2.01e-6 | 1.08e-5 | (4) |
| Artery (coronary) | Purine-containing compound biosynthetic process (BP) | 0.117 | 2.48e-6 | 7.52e-6 | (4) |
| Artery (tibial) | Microtubule-based movement (BP) | 0.582 | 2.45e-5 | 2.20e-3 | |
| Artery (tibial) | Microtubule binding (MF) | 0.525 | 2.34e-5 | 3.41e-3 | |
| Heart (atrial appendage) | Regulation of blood pressure (BP) | 0.000777 | 2.77e-9 | 2.04e-11 | (5) |
| Heart (atrial appendage) | Vascular process in circulatory system (BP) | 5.55e-6 | 4.91e-10 | 1.37e-9 | (5) |
| **PP** | | | | | |
| Artery (coronary) | Nucleoside phosphate biosynthetic process (BP) | 0.500 | 1.02e-5 | 2.89e-4 | (6) |
| Artery (coronary) | Purine-containing compound biosynthetic process (BP) | 0.205 | 2.44e-5 | 3.63e-4 | (6) |
| Artery (tibial) | miRNA-145 targets | 0.638 | 1.01e-5 | 6.20e-7 | |
| Heart (atrial appendage) | Cellular response to nitrogen compound (BP) | 0.0167 | 2.18e-5 | 8.10e-7 | |
| Uterus | Development of primary sexual characteristics (BP) | 0.0144 | 7.56e-7 | 2.84e-7 | (7) |
| Uterus | Sex differentiation (BP) | 0.00523 | 6.26e-7 | 4.28e-7 | (7) |

Marginal gene-set *p*-values are from step 2 of the analysis workflow, after correcting for general confounders. The 'top 25%' *p*-values are for the gene set of the 25% genes with the highest residual expression on the tissue, conditioned on the whole set and the tissue expression. Interactions that likely reflect a single shared association are marked by the same number in the 'Shared' column
*SBP* systolic blood pressure, *DBP* diastolic blood pressure, *PP* pulse pressure, *BP* biological process, *MF* molecular function

## Discussion

The development of the analysis workflow presented in this paper was motivated by the problem of correlated gene properties, and the confounding and the multiplicity of redundant overlapping associations that could result from this. The results from the blood pressure analyses show that this can indeed present a serious problem in practice.

General confounding factors, here primarily the involvement of overall and tissue-specific expression, are shown capable of inducing significant associations in a large number of gene properties. Those gene properties subsequently also overlap with and confound each other, with a subset of the significant gene properties accounting for the associations of the rest as well as for large amounts of sub-significant associations in the other gene properties.

Correcting for these issues drastically reduces the number of gene-property associations, which implies that traditional GSA lacking such corrections is liable to yield large numbers of associations which are likely not biologically relevant to the phenotype. Conclusions drawn from such analyses are therefore at considerable risk of being incorrect, and potentially very misleading. These same issues most likely affect other, similar types of analysis as well, such as network analysis or SNP-set analysis.

Our extension to interaction GSA opens up new avenues of analysis. Results for the blood pressure phenotypes suggest that there may be numerous signals in the annotation that a standard GSA cannot reliably detect, if it can detect them at all. This is perhaps best exemplified by the regulation of blood pressure gene set. Based on its marginal associations there is little evidence that it is involved in blood pressure genetics, and would not have been found in a traditional GSA. Yet it has a very strong interaction with heart-specific expression for both SBP and DBP, and the subset of top expressed genes in the set is highly associated. This same pattern is found for many of the tissue expression by gene-set interactions, with many of those gene sets having entirely unremarkable marginal p-values. The same is suggested by the exploratory interaction analysis, with negative interactions in particular seen to mask strong associations.

Taken together, our results thus show that a traditional GSA is doubly vulnerable. Firstly, due to confounding many marginal associations are likely to be found that are biologically irrelevant, or the byproduct of more specific interactions. This can lead to potentially very misleading conclusions, and wasted effort trying to follow them up. Secondly, many gene properties may only affect the phenotype in combination with other gene properties, rather than on their own. Marginal associations for such gene properties will often be weak or absent altogether, and therefore unlikely to be found in traditional GSA. Our extended GSA approach can address these issues, pruning away many likely irrelevant associations through conditional analysis and detecting novel additional or more refined signals with the interaction model.

**Table 4 Significant and retained interactions from post hoc gene set by gene set interaction analysis for pulse pressure**

| Gene-set interaction pair | Size | Overlap | Marginal p-value | Full model p-values | | Shared |
|---|---|---|---|---|---|---|
| | | | | **Main** | **Interaction** | |
| Cardiovascular system development (BP)[a] | 764 | 100 | 1.79e-9 | 2.74e-5 | 1.44e-5 | (1) |
| × Chemical homoeostasis (BP) | 844 | | 0.000672 | 0.0836 | | |
| Cardiovascular system development (BP)[a] | 764 | 147 | 1.79e-9 | 0.000210 | 2.01e-5 | (1) |
| × Homoeostatic process (BP) | 1277 | | 0.000981 | 0.100 | | |
| Cell adhesion molecule binding (MF) | 180 | 21 | 1.28e-6 | 0.000574 | 0.000751 | |
| × Glycosaminoglycan binding (MF) | 201 | | 0.0904 | 0.602 | | |
| Cell proliferation (BP) | 645 | 204 | 1.60e-8 | 0.00311 | 0.000204 | |
| × Regulation of intracellular signal transduction (BP) | 1581 | | 0.0197 | 0.340 | | |
| Cell proliferation (BP) | 645 | 74 | 1.60e-8 | 5.51e-5 | 7.62e-6 | |
| × Regulation of intracellular signal transduction (BP) | 596 | | 0.454 | 0.958 | | |
| Regulation of transcription from RNA polymerase II promotor (BP) | 1682 | 203 | 9.97e-7 | 0.00127 | 1.73e-5 | |
| × Homoeostatic process (BP) | 1277 | | 0.000981 | 0.125 | | |

Marginal gene-set p-values are from step 2 of the analysis workflow, after correcting for general confounders. Interactions that likely reflect a single shared association are marked by the same number in the 'Shared' column
BP biological process, MF molecular function
[a] The same interaction exists for circulatory system development, which is identical to cardiovascular system development and therefore omitted from this table

Aside from demonstrating the utility of our proposed analysis workflow, our analyses also provide a variety of insights into the genetics of blood pressure, and many of the individual associations fit well with the existing blood pressure literature. The tissue expression analyses detected associations for several cardiovascular tissues, which are highly adapted to blood pressure fluctuations. In the cellular component domain constituents of the (sacromeric) cytoskeleton, including actin and T-tubules, were identified[18], and the majority of detected biological processes are involved in blood vessel and heart formation. These include cardiovascular and circulatory system development, cardiocyte differentiation, SMC regulation and (cardiac) mesenchyme development.

The interaction analyses provided further detail for these associations. Expression in the heart (atrial appendage) interacted strongly with the regulation of blood pressure gene set for both SBP and DBP, which possibly reflects the role of atrial natriuretic peptide in the homoeostasis of sodium and water retention[19]. This is supported by the interactions of cardiovascular system development with homoeostatic processes for SBP and PP. Heart expression also interacted with cellular response to nitrogen compound for PP, which fits the known natriuretic peptide–nitric oxide pathway and guanylate cyclase signalling systems that are targeted by nitroglyceride[20].

Artery tissues were found to exhibit interactions with nucleoside phosphate and purine-containing compound biosynthetic process for SBP, DBP and PP. Nucleoside and purine are not only constituents of RNA and DNA but are also involved in metabolic processes such as signal transduction and regulation of enzyme activity[21]. This therefore aligns with the interactions found between cell proliferation and regulation of intracellular transport and signal transduction for PP, supporting the role of purinergic signalling in the proliferation of vascular smooth muscle and endothelial cells[22].

Further evidence for a role of signal transduction was found in the associations of nitric oxide and cGMP for DBP. Nitric oxide is an important signalling molecule that regulates vascular tone by acting as a vasodilator via the cGMP signalling cascade and intracellular $Ca^{2+}$ levels[23,24]. Also found for DBP was reactive oxygen species biosynthesis, which has been implicated with cardiovascular disease including hypertension[25].

The miRNA target genes, which regulate various physiological and pathophysiological processes at a post-transcriptional level[26], were associated for all three blood pressure phenotypes. Although none of the individual miRNA target sets was significant, an interaction was found between tibial artery expression and the miRNA-145 target set. This interaction can be explained by the influence of miRNA-145 on differentiation[27] and phenotype switching of vascular SMCs[28,29], and the upregulation of miRNA-145 in endothelial cells in response to shear stress and hypertension[30].

No associations were found for the kidney cortex or the adrenal gland in the tissue expression analysis, which is surprising considering the regulatory role of the renin–angiotensin–aldosterone system on blood volume and systemic vascular resistance[31] and known associations of renal sodium regulatory genes variants with blood pressure[32]. One possible explanation is that the available kidney cortex expression is too general. It has been shown that unique and highly distinctive patterns of gene expression exist for glomeruli, cortex, medulla, papillary tips and pelvic tissue[33], and associations with blood pressure genetics may only exist in such more specific tissues. Regulation of urine volume was also found to be associated with both SBP and DBP, which supports the hypothesis that kidney involvement may be quite specific.

Also notable were the associations of several female reproductive organ tissues, most prominently the uterus, for all three phenotypes. This may point to the involvement of an underlying hormonal pathway, correlated to ovarian expression. Such a pathway could reflect the known protective effects of oestrogens on cardiovascular disease and hypertension[34]. Alternatively, expression in these tissues may serve as a proxy for placental expression, which is not available in the GTEx data. The placenta has been shown to play a role in blood pressure regulation during pregnancy[35], and placental functioning is directly related to fetal growth which has been linked to the development of hypertension during adult-life of the child[36,37].

The application of traditional GSA has previously led to novel biological hypotheses on human physiology and the pathophysiology of disease, and the GSA presented in this paper improves on that promise for blood pressure phenotypes. Our results, filtered and refined using the extended analysis workflow, suggest a variety of possible avenues by which the role of genetics in blood pressure may be explained. Exploring these avenues could advance our understanding of blood pressure and the identification of therapeutic targets for cardiovascular disease, and our extended analysis can be used generally to provide the same for other phenotypes as well.

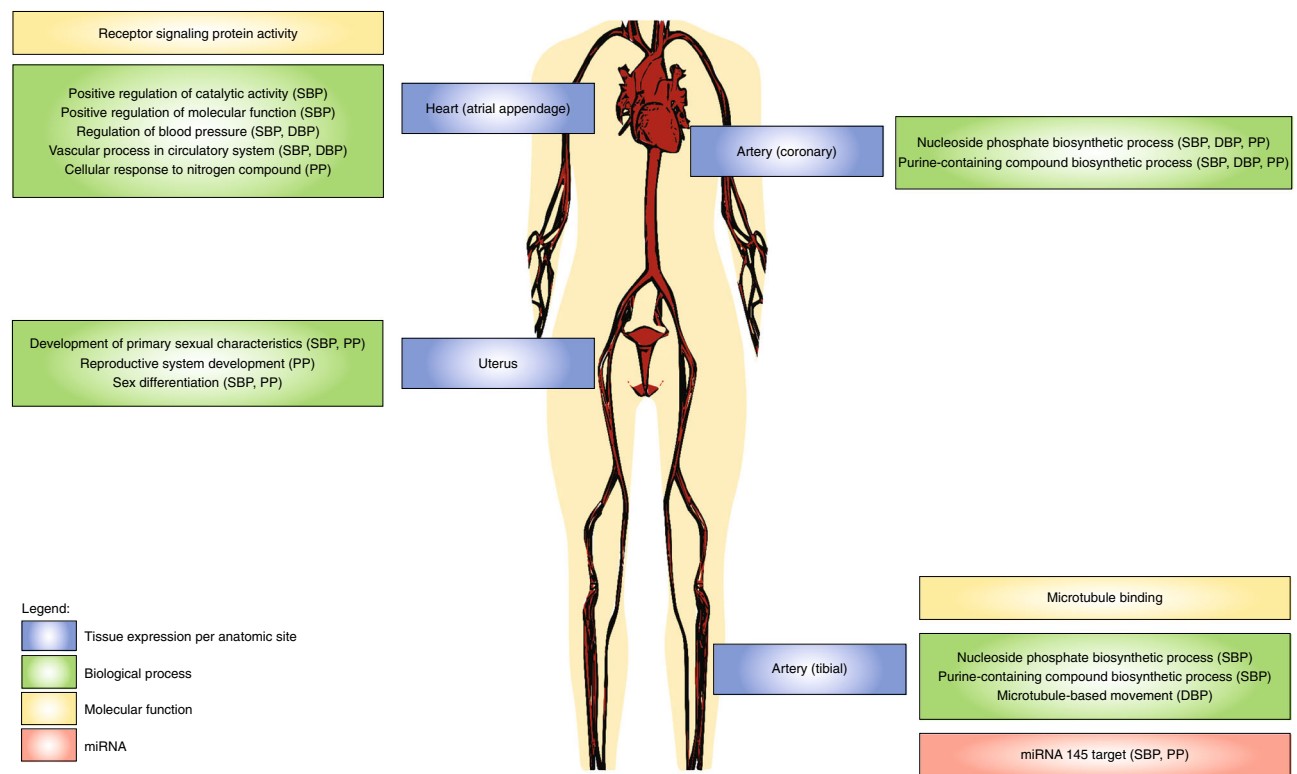

**Fig. 4** Illustration of results from post hoc tissue expression by gene set interaction analysis. Only interactions that were initially significant and retained after follow-up checks are shown. Tissue expression per anatomic site (blue), biological process (green), molecular function (yellow) and miRNA (red)

## Methods

**Core GSA framework**. We use GSA implemented in MAGMA (v1.07), a detailed description of which can be found in De Leeuw et al.[5]. Briefly, the model is based on a linear regression framework with genes as data points, with the regression equation $Z=\beta_0+B\beta_B+S\beta_S+\varepsilon$, with $\varepsilon \sim \mathrm{MVN}\left(0, \sigma_\varepsilon^2 \hat{\Sigma}\right)$. Gene $p$-values $P_g$ are first computed from the SNP data for each gene $g$. These are transformed to $Z$-scores, $Z_g = \Phi(1 - P_g)$ with $\Phi$ the probit function, such that higher $Z_g$ correspond to stronger genetic associations with the phenotype. The gene set is encoded in the variable $S$, with $S_g=1$ if gene $g$ is in the gene set and $S_g=0$ otherwise.

Linkage disequilibrium (LD) between genes is quantified in the gene–gene correlation matrix $\hat{\Sigma}$, which is scaled by the variance $\sigma_\varepsilon^2$ to model the residuals. Several common technical confounders are included as covariates, represented by the matrix $B$ in the regression. These are: the number of variants in each gene, an estimate of the LD within each gene, the inverse of the mean minor allele count of variants in each gene, and the sample size on which each gene $p$-value is based. For each of these variables, the log transformation of the variable is also included as a covariate.

A one-sided test is performed on the coefficient $\beta_s$ of the null hypothesis $\beta_s=0$ against the alternative $\beta_s>0$, testing whether the genes in the gene set are more strongly associated with the phenotype than other genes. This constitutes a competitive test (see De Leeuw et al.[1] for a discussion on key differences with self-contained GSA).

The model can also analyse non-binary gene properties, such as gene expression. In this case $S$ is a continuous variable, and the coefficient $\beta_s$ reflects the degree to which the genetic association of a gene changes as the value for the tested variable increases. By default, a two-sided test is performed on $\beta_s$ when analysing continuous gene properties since, in contrast to gene sets, negative associations may be informative as well.

Throughout the text, we use 'gene property' to refer to any type of gene-level variable, and 'gene set' to refer specifically to a binary gene property.

**Conditional and interaction GSA model**. Conditional and interaction GSA is implemented by generalising the core regression framework. For conditional analysis a matrix of additional covariates $C$ is included in the model, to obtain $Z=\beta_0+B\beta_B+C\beta_C+S\beta_S+\varepsilon$. The $\beta_S$ now reflects the conditional effect of $S$ on the genetic association $Z$, corrected for the effects that the covariates in $C$ have on $Z$.

For the interaction GSA an interaction term $S_{12}$ is defined as the product of two gene properties $S_1$ and $S_2$, with $S_{12_g} = S_{1_g} \times S_{2_g}$. Then $S_{12}$ is tested conditional on $S_1$ and $S_2$ to determine if there is any interaction between them, in the model

$Z=\beta_0+B\beta_B+S_1\beta_1+S_2\beta_2+S_{12}\beta_{12}+\varepsilon$. The test can be either two-sided or one-sided in either direction.

An interaction of this type means that genes that have high values for both gene properties are more strongly (or weakly, if $\beta_{12}$ is negative) associated with the phenotype than genes that have high values for only one of the two. This suggests a specific role for that combination of properties. This role may be limited to that combination, but can also be in addition to significant main effects ($\beta_1$ and $\beta_2$) of the gene properties. For pairs of gene sets, $S_{12}$ simply corresponds to the set of genes included in both gene sets.

The conditional and interaction GSA models are implemented in MAGMA as part of the GSA framework, and can be used with any of the gene analysis models available in MAGMA. It can therefore be applied to both raw genotype data as well as SNP summary statistics from any type of single variant analysis.

**Analysis workflow**. The extend GSA workflow consists of six analysis steps (Fig. 2). An initial GSA is first performed to select significant gene properties, and the subsequent steps are then used to provide further information on their associations. This is then used to aid interpretation of the results, and to discard likely irrelevant gene properties from consideration. It can also flag some gene properties as requiring further analysis and data before interpreting them, if the evidence for their biological relevance to the phenotype is ambivalent. The initial GSA results are thus progressively refined, improving the reliability of the conclusions that can be drawn. An overview of the six steps is provided here. An extensive guideline on performing the analyses and interpreting the results can be found in the Supplementary Methods.

The first step of the analysis workflow is a standard MAGMA GSA (with only the automatic correction for technical confounders). Only gene properties significant in this GSA are directly evaluated in the subsequent steps (except step 6). In the second step, the significant gene properties are conditioned on likely confounders, and the impact those confounders have on their associations is assessed. Gene properties that are no longer significant at the significance threshold used in step 1 are then discarded.

In step 3, remaining significant gene properties are conditioned on each other. This helps determine the extent to which their associations overlap, and to identify which of those associations are most likely relevant for the phenotype. Gene properties are selected in a stepwise fashion on the strength of their associations and the way those associations overlap with each other. In each selection step, gene properties are conditioned on both the gene properties already selected and the general confounders from the second analysis step. Gene properties for which the association is largely or wholly explained by other gene properties are discarded;

gene properties which are found to share a single underlying association that cannot be disentangled are selected and interpreted jointly.

The fourth step applies only to gene sets, and checks for outliers and signs of confounding effects not detected in the previous steps. For each gene set QQ-plots of the residual $Z$-scores of genes in the set are created, adding a confidence band to visualise the degree of deviation expected by chance. These are inspected for signs that the association of the gene set may be driven by a smaller subset of genes in the set, indicating possible confounding. If not uncovered in the post hoc interaction analyses, the source of confounding could then be investigated further using targeted analyses with additional data or annotation. If the likely associated subset is very small the problem is likely one of outliers instead, and the gene set can be discarded altogether.

In the fifth step, interaction analyses are performed for all the remaining significant gene properties. This can narrow down the significant associations to more specific effects that occur only in combination with other gene properties. Positive interactions are tested with all other available gene properties; for interactions between gene sets, this is restricted to pairs of gene sets for which the overlap between the sets is not too large or small, as otherwise the interaction term is not meaningfully defined.

In the optional sixth step, an exploratory interaction analysis is performed in order to detect additional interactions. An initial list of gene properties is generated based on their marginal associations, and interactions with all other gene properties are tested for this list as in step 5. A liberal selection criterion such as FDR-controlled significance is recommended for creating the initial list. In contrast to step 5, two-sided tests are performed for the interactions. This allows for the detection of negative interactions, which would point to involvement in the phenotype of a particular gene property only in the absence of another gene property. This step is independent of the previous steps, and therefore requires separate multiple testing correction.

**Genotype and phenotype data**. Primary quality control and imputation of the UK Biobank (July 2017 release) data was performed by UK Biobank itself[7]. We applied additional QC and filtering of variants and individuals to obtain a sample of independent individuals of European ancestry, containing hard-called genotypes with MAF greater than 0.000001 and missingness of at most 5%. Since poorly imputed SNPs can bias the results, only variants of high imputation quality (info score of at least 0.9, variants imputed on HRC panel only) were included in the analysis. Full details on the data and QC can be found in the Supplementary Methods. The processed data set used for the blood pressure analyses contained 360,243 individuals and 13,923,638 autosomal variants.

In our analyses, three phenotypes were analysed: SBP, DBP and PP. SBP and DBP were corrected for use of blood pressure-lowering medication, adding 10 and 15 mm Hg respectively to the measured values for individuals known to use such medication[38]. PP was computed as PP = SBP−DBP. Thirty principal components were included as covariates to correct for population structure in the data, computed using FlashPCA[39]. Other covariates included in the analysis were sex, age, age squared, BMI, Townsend Deprivation index, and genotyping array indicator.

To further validate the results from the UK Biobank analysis, a replication analysis was performed using the 2011 ICPB GWAS data[40]. Details for this replication analysis can be found in the Supplementary Methods.

**Annotation**. Variants were annotated to genes based on NCBI (37.3) gene definitions[41], mapping variants to a gene if they were located in the transcription region of that gene, or within two kilobase upstream or one kilobase downstream of the transcription region. A total of 18,285 autosomal protein-coding genes had at least one variant mapped to them, and 43.7% of the variants in the data mapped to at least one gene. Variants not mapped to any gene were not used in the analysis.

Gene annotation from five different domains was used in the analysis: tissue-specific gene expression data, three Gene Ontology domains, and miRNA target sets. Gene Ontology and miRNA target gene sets were obtained from MsigDB (v6.0)[8]. For the miRNA target sets, an additional gene set of all genes contained in at least one of the target sets was created, reflecting general miRNA target status.

GTEx (v7)[9] was used for the gene expression data. Mean RPKM values were computed across gene and tissue. These were truncated down to 50, incremented by one, then log-transformed to obtain a per-tissue expression score. Average scores across tissues were computed as a measure of the overall expression level of each gene. Ensembl gene IDs were mapped to Entrez IDs for the genes in the data, resulting in expression scores for 17,064 genes in the data.

**Simulation study**. A random subsample of 10,000 individuals was taken from the UK Biobank data, filtering variants with MAF smaller than 1% and variants not mapped to any gene. Continuous phenotypes were simulated for this data by constructing a genetic component and adding normally distributed noise such that the genetic component explained 10% of the phenotypic variance. The genetic components were created by designating 1000 genes as causal, then selecting a subset of SNPs from each of these genes as effect SNPs and combining them (see Supplementary Methods for full details).

Simulated phenotypes were analysed in PLINK 1.9 (ref. [42]) to obtain SNP $p$-values. Ten genetic components were constructed (designating new causal genes

and SNPs), with 100 replicates for each. Multiple phenotypes with new random noise were generated for each replicate, using meta-analysis on the SNP $p$-values to obtain GWAS results representing sample sizes of 10,000, 50,000, and 100,000.

Pairs of overlapping gene sets were then constructed, containing different patterns and proportions of causal genes. In each condition an initial gene set was created containing a specified proportion of causal genes. Another gene set was then created overlapping with it, as either a subset, a superset, or partially overlapping set. Genes in the overlap were randomly selected from the initial gene set, with the rest randomly sampled from the remaining genes. For evaluation of the interaction model, only partial overlap conditions were used.

Additional parameters that were varied across conditions were the gene set sizes, the degree of overlap, and the level of association assigned to the initial gene set. For the interaction model, the level of main effect association assigned to the second gene set was also varied. A full description of the simulation settings and results is given in the Supplementary Methods.

In each condition, ten gene sets overlapping with the initial set were created. For the conditional model simulations the marginal association and association conditional on the initial set were tested. For the interaction model, the interaction term was tested either as a gene set by itself or using the interaction model. Results were aggregated per condition over the ten sets and the 1000 GWAS replicates, computing type 1 error rates at different significance thresholds.

**Primary GSA**. Analyses were performed using MAGMA (v1.07)[5]. Phenotypes were first regressed on the covariates, using the resulting residuals as input for the MAGMA gene analysis. The SNP-wise (multi) model was used for the gene analysis. This model combines the SNP-wise (mean) model (more sensitive to many smaller SNP associations in a gene) and the SNP-wise (top) model (more sensitive to a single large SNP association in the gene) to obtain a good distribution of power over different genetic architectures. This model is recommended when the number of SNPs in a data set is very large, as the SNP-wise (mean) and PC regression models are less sensitive to detecting gene associations when a single strong SNP effect is present in a gene containing many other SNPs.

To deal with rare variants, per gene SNPs with a minor allele count smaller than 100 were aggregated into a weighted burden score. This was then included in the model in the same way as normal SNPs, replacing the rare variants. At most 25 SNPs were used per burden score. For genes with more than 25 rare variants, multiple burden scores were created.

All GSA was performed using this gene analysis output. Bonferroni correction was used to correct for multiple testing, separately for each phenotype. It was also applied separately for each domain, corrected for the number of domains, for a significance threshold of $\alpha_D = \frac{0.05}{5 \times K_D} = \frac{0.01}{K_D}$ per domain $D$ with $K_D$ the number of tests for that domain. In all the analyses one-sided tests were used, testing for positive associations.

**Conditional GSA**. After the initial GSA, analyses were repeated conditioning on potential general confounders. Overall gene expression was included for all domains. For the four gene set domains, tissue-specific expression for coronary artery, tibial artery, heart (atrial appendage), and uterus were also conditioned on; the miRNA target set analyses were additionally conditioned on general miRNA target status. For conditional analyses of the gene sets, missing tissue expression values were set to the median expression value for that tissue. Only gene sets and tissues still significant at the original threshold were retained.

Conditional analyses were then performed to evaluate overlap between associations of significant associations. The stepwise procedure was used per domain for the significant and retained gene properties until there were no remaining associations with conditional $p$-values below 0.05 (see Analysis workflow above and Detailed overview of blood pressure analysis in the Supplementary Methods). For gene sets, associations retained after this selection were then also conditioned on those from the other domains. All these analyses also included the general confounders as covariates. After this set-specific QQ-plots were created for all retained gene sets to inspect them for signs of outliers and hidden confounding.

**Expression by gene set interaction analysis**. After the conditional analyses, post hoc interaction analyses were performed for the top tissue expression levels. Genes with no expression values were removed, and interactions were then tested with all gene sets of at least 100 genes. To make the results more comparable across phenotypes, the same tissues were used for all three phenotypes, testing interactions for coronary and tibial artery, heart (atrial appendage) and uterus, as well as for overall gene expression.

For each tissue, overall expression and its interaction with the tested gene set were included as covariates. For miRNA target sets, general miRNA target status and its interactions with overall expression and the tissue expression were additionally included. One-sided tests were performed for the interaction terms, testing for positive interactions. Bonferroni correction was performed per tissue, correcting for the 1495 interactions tested per tissue.

To check for outliers, scatterplots of residual tissue expression (corrected for the overall expression) by gene $Z$-scores were created for all significant interactions. Each plot only used genes in the set, and both variables were normalised within

those genes. Genes were marked as an outlier if they were more than two standard deviations from the origin and all genes within two standard deviations were either further from the origin or themselves marked as outlier.

The analysis was then repeated with the marked outliers removed from the gene set. A gene set was also constructed of the top 25% residually expressed genes in the gene set (excluding outliers), in which was then tested conditional on the whole gene set. Interactions for which neither follow-up test was significant were discarded.

**Gene set by gene set interaction analysis**. Post hoc interaction analyses were also performed for all significant and retained gene sets, testing interactions with other gene sets. Interactions were only tested for gene-set pairs if there was meaningful overlap between the gene sets: for each set in the pair, the overlap with the other gene set as well as the part not overlapping with the other gene set was required to be at least 20 genes, and at least 10% of the genes in the gene set. One-sided tests for positive interactions were performed, conditioning on the general confounders. Bonferroni correction was applied separately for each of the significant and retained gene sets, correcting for the number of interactions tested for that gene set.

An exploratory interaction analysis was also performed. Gene sets were selected using FDR correction (Benjamini–Hochberg, at $\alpha = 0.05$), separately for each of the four gene-set domains. For each of these gene sets, interactions were tested with all other gene sets for which there was meaningful overlap, using the same criteria as in the post hoc interaction analysis. Two-sided tests were performed on the interactions, conditioning on the general confounders. Bonferroni correction was applied for the total number of interactions tested.

**Code availability**. The MAGMA analysis software can be obtained for Linux, Windows and Mac platforms from http://ctg.cncr.nl/software/magma.

## Data availability

The raw genotype and phenotype data analysed in this study were used under license from UK Biobank (http://www.ukbiobank.ac.uk), and restrictions apply to its availability. However the data are available from the authors upon reasonable request, if permission is given by UK Biobank.

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

## Acknowledgements

This work was funded by The Netherlands Organization for Scientific Research (NWO VICI 453-14-005, 645-000-003). The analyses were carried out on the Genetic Cluster Computer, which is financed by the Netherlands Scientific Organization (NWO: 480-05-

003), by the VU University, Amsterdam, The Netherlands, and by the Dutch Brain Foundation, and is hosted by the Dutch National Computing and Networking Services SurfSARA. This research has been conducted using the UK Biobank Resource under project 16406. We thank the participants and researchers who collected and contributed to the data.

## Author contributions

C.dL., T.H. and D.P. conceived of the study. C.dL. developed the statistical method and performed the analyses. S.S. prepared the UK Biobank data for analysis. C.dL., I.A.D. and D.P. wrote the paper. All authors discussed the results and commented on the paper.

## Additional information

**Competing interests:** The authors declare no competing interests.

