## [Peer Review File · Nature Communications]

Reviewer #1 (Remarks to the Author):

In this paper, the authors extend their previous gene level or gene set analysis method (MAGMA) for conditional and interaction analysis. They propose a streamlined pipeline to minimize false positive outcome. They apply the method to UK biobank blood pressure phenotypes and find interesting gene set or interaction results.

Overall, the paper is well written. The extension pipeline of MAGMA is not super unique or creative, but sounds reasonable, partly because the MAGMA framework is already quite general and extendable. They have done a really careful job to control for possible confounding, which is great. The pathway terms they found for blood pressure make sense (e.g. cardiovascular system).

They have done a great job in describing each step of the pipeline and explaining why it was needed. Seeing that many method papers nowadays are not clear or kind, I feel that the way this paper is written can be a good practice to follow.

I have a few minor comments.

1. In their MAGMA paper, the main method they advertised for obtaining a gene-level p-value was based on F-test on principal components. It seems that in this paper, they used a p-value based version where they mix the “mean” version and the “top” version. Was there any reason for this choice – if they recommend this, the readers and users should know why they are supposed to use this version.
2. It is unclear how they controlled for FDR. Is it Benjamini-Hochberg method?
3. The ultimate goal can be to allow users to use this pipeline in their studies. To do this, a careful step-by-step description of the pipeline (or manual) should be available. Moreover, it will be great if as many parts of the pipeline are automated. For example, you can automate the generation of QQ plot and give outputs of how much each QQ is deviated? (e.g. like genomic-control deviation factor).
4. Sup Figure 2B is missing?

Reviewer #2 (Remarks to the Author):

The authors previously developed a tool MAGMA for testing gene set enrichment for genomic studies including genome-wide association studies. In current manuscript, they extended the framework to conditional analysis and interaction analysis. The primary application of the tool is to better understand which gene set is truly associated with the trait when multiple gene sets are overlapping. While it is potentially useful for GWAS secondary analysis, I have several comments.

(1) One of the claim of the manuscript is that the conditional approach has the potential to discovery “causal” enrichment. This is misleading. The causality based on mediation analysis typically requires explicit biological hypotheses. For example, SNP A is associated with lung cancer risk and smoking; smoking is associated with lung cancer risk. A mediation analysis can be done using conditional analysis to establish the causality of the SNP A. For current manuscript, the conditional analysis is including two gene sets with overlapping genes. I do not see the way to make any claim about causality. While the conditional analysis helps reduces the number of associations, the author need to think of another way to interpret the results and the benefits of this analysis. Need to clearly define “causality” and “spurious findings” in this setting.

(2) Type-I error need to be checked. The authors have applied stepwise analysis (1) standard GSA for marginal association (2) for significant associations perform conditional and interaction analysis. Simulations are required to access the type I error rate for the whole procedure.

(3) As a methodology paper, large scale simulations are important to understand the statistical property of the test, which includes but not limited to (A) the way of summary gene-level Z-score based on SNP PCA analysis (is this approach powerful?), (B) when gene set A is selected if A and B are overlapping and are both associated (enriched) with the trait and (C) interaction testing.

(4) Application to blood pressure reduced the number of significant enrichment. How do we know the new results are scientifically more plausible? For these significant results, replication study is preferred in an independent GWAS data set.

(5) While MAGMA made it clear to be applicable to GWAS summary data, the authors did not mention whether the current approach can be used to GWAS summary data. If this analysis cannot be done based on GWAS summary data, the impact will be limited.

Minor comment:

You have two “Figure 1”.

Reviewer #3 (Remarks to the Author):

The authors provide a new extended approach to gene-set analysis, by incorporating conditional and interaction analyses, in contrast to the traditional GSA approach which only assesses the marginal associations of gene properties. This enables the identification of confounding as well as the discovery of novel gene properties that would not be detectable marginally, altogether improving the robustness, reliability and specificity of the results and conclusions. And the results can be restricted only to those which are more likely to be causal. This is good work, well done.

This approach is applied to blood pressure phenotypes from the UK Biobank data, as an example, considering tissue expression data from GTEx, miRNA target data and Gene Ontology data. They illustrate that many of the significant results from traditional GSA would lead to misleading conclusions for gene properties which were not actually likely to be causally relevant, or that associations were not tissue-specific. The refined conclusions for blood pressure were interesting, for example with the ability to show distinct differences between the three different blood pressure phenotypes, which are not usually revealed through BP GWAS papers.

Major comments:

1) Whilst I agree with the authors that “the analyses of blood pressure phenotypes provide novel insights into their genetic etiology”, I think the paper is lacking an initial literature review of what has been reported so far in this context from blood pressure genetic analyses. I was surprised that none of the recent BP GWAS papers were cited within the references. In fact, the reference list seems somewhat short. Many recent BP GWAS papers have performed pathway analyses, expression and enrichment analyses as secondary analyses to their GWAS analyses. It would therefore be helpful, for the readers and research community, if the authors could briefly compare the GSA approach to other such approaches, e.g. IPA or DEPICT, etc, from a methodological perspective. Then, also, contrast the final conclusions here on significant gene properties, with those that have been previously reported in recent BP genetics papers. The authors here do compare their refined results to what would have been obtained from traditional GSA alone, but only within the

context of their own analysis of the UKB data. It would also be interesting to compare to what was already present in the literature.

2) In the Online Methods we realise that this analysis was only performed using the 2015 interim N~150k release of the UKB data. As many submitted journal papers are now using the full July 2017 N~500k UKB data, I think that it needs to be made clearer at the start of the paper, that this only uses the interim data. Ideally, time permitting, could all analyses be updated now to the full N~500k UKB data? Or at least, please acknowledge this in the discussion, and perhaps comment on how the authors expect the accuracy of results or final conclusions may or may not change with the increased sample size and newly imputed genetic data?

3) Please provide a more thorough detailed description of your own QC of the UKB data. The opening paragraph of the Supplementary text is still very brief. For example:

a. What method / criteria did you use to remove non-European individuals?

b. What degree of relatedness did you use for exclusions of related individuals?

c. Please state the exact number of samples excluded according to each individual exclusion criteria separately? E.g. N for non-Europeans, relatives, sex-discordant, etc?

d. I note that the INFO threshold of 0.8 is much stricter than the imputation quality cut-offs usually used in GWAS. Please comment and explain why a stricter threshold was used for GSA here?

4) From the Supplementary text, I note that data was converted to best-guess genotype format PLINK data for analysis. Please comment and explain why imputed dosage data cannot be used within these GSA analyses? And if not, please discuss this as a possible limitation?

5) The three most highly implicated tissues for BP remaining seem to be the heart, coronary artery and uterus. Please could you provide further discussion on the biological interpretation of these final findings? For example, whilst the first two may be less surprising for blood pressure as a cardiovascular phenotype, perhaps you could offer an interpretation of the relevance of the uterus tissue expression?

6) The authors used MAGMA software for the initial GSA. Have these conditional and interaction extensions of the GSA framework to the software also been made publicly available as software tools, to enable other analysts to use this proposed analytical approach?

Minor comments:

1) On pg.5 you say that additional analysis and follow-up research would still be required for the other four sets. Please could you discuss more details of what these additional analyses would be?

2) I note an error in the Figure 1 legend, where left and right are the wrong way around.

3) In Figure 2, please state the significance thresholds used (i.e. in addition to stating within the text).

- 4) Pg.14 shows a third Figure, which is labelled as "Figure 1", so I am assuming this should be titled Figure 3?
- 5) On pg.20 the authors helpfully give warnings regarding the analysis of "small gene sets" which are "particularly vulnerable". Please could you give more detail as to how small is "small"?
- 6) Please state the MAF variant filter in the Online Methods too, not only in the Supplementary file, so that the reader can see where the ~21M variants have come from. Furthermore, as the MAF threshold was 0.1%, please could you discuss any implications of using rare variants within these GSA analyses?
- 7) It then becomes confusing that even though ~21M variants are stated to be in the dataset (pg. 21), that then only ~9M variants can actually be annotated to genes (pg. 22). Does this mean that the GSA analyses then only actually consider these final ~9M variants? Please clarify.
- 8) Please justify the covariates that you have used? (pg. 21) I note that BMI is not adjusted for, even though this covariate is usually used within most BP GWAS analyses?
- 9) For any readers who are less familiar with MAGMA, please also state all the criteria used, even though all the settings were at "default values" (pg. 23)

OVERVIEW OF CHANGES

We would like to thank the reviewers for their extensive and constructive comments, and have endeavoured to address all of them in our revision. A point-by-point reply to all the comments is provided below; reviewer comments are in *italics*, with our reply in **bold**. Changes in the manuscript file have also been highlighted in blue.

A general overview of the most significant changes made to the manuscript is also provided here. The largest change results from switching to the full UK Biobank (July 2017) release, which replaces all the results from the original manuscript. With the required reanalysis, a number of accompanying changes have been made to the rest of the data and analysis: fifteen additional PCs as well as BMI were added as covariates in the analysis of the genotype data; the GTEx gene expression data has been updated to version 7, removing the tissue-specific eQTL analysis in the process as it did not add anything to the tissue-specific gene expression analysis; the analysis workflow has been refined in several places, providing more explicit recommendations for thresholds and cut-offs, improving the set-specific QQ-plots, and switching from FDR correction to the stricter but more robust Bonferroni FWER correction.

The manuscript has been extensively rewritten with the results of the reanalysis as well as to fit the Nature Communications article format. More background and biological interpretation have been added to the Introduction and Discussion. Several parts of the Methods section have been moved to the Supplemental Note and expanded into an analysis guideline for running the analysis and interpreting the results. Subsequently, some paragraphs about method and analysis were moved from the Supplemental Note to the Methods to provide a more comprehensive description there. A simulation study has been added, with corresponding sections in Results, Methods and Supplemental Note; a replication analysis was also conducted, described in the Supplemental Note.

REVIEWER 1

In this paper, the authors extend their previous gene level or gene set analysis method (MAGMA) for conditional and interaction analysis. They propose a streamlined pipeline to minimize false positive outcome. They apply the method to UK biobank blood pressure phenotypes and find interesting gene set or interaction results.

Overall, the paper is well written. The extension pipeline of MAGMA is not super unique or creative, but sounds reasonable, partly because the MAGMA framework is already quite general and extendable. They have done a really careful job to control for possible confounding, which is great. The pathway terms they found for blood pressure make sense (e.g. cardio vascular system).

They have done a great job in describing each step of the pipeline and explaining why it was needed. Seeing that many method papers nowadays are not clear or kind, I feel that the way this paper is written can be a good practice to follow.

I have a few minor comments.

1. In their MAGMA paper, the main method they advertised for obtaining a gene-level p-value was based on F-test on principal components. It seems that in this paper, they used a p-value based version where they mix the "mean" version and the "top" version. Was there any reason for this choice – if they recommend this, the readers and users should know why they are supposed to use this version.

The SNP-wise (mean) and PC regression are well-suited for gene effects based on multiple weaker SNP effects. But they are less sensitive to gene effects based on a single strong SNP effect embedded in a large gene (ie. containing many SNPs), especially if the effect has a lower MAF (and

therefore generally lower LD with neighbouring SNPs). The SNP-wise (top) model has essentially the opposite pattern of sensitivity. Combining the SNP-wise (top) model with one of the other two, as the SNP-wise (multi) model does, therefore helps ensure that the results are not biased towards gene associations of one or the other kind. This is especially pertinent in data that contain many SNPs and include lower MAF ranges such as the UK Biobank data, hence why it was used.

We have added this as an explicit recommendation in the Methods section, in the subsection *Primary gene-set analysis*.

2. *It is unclear how they controlled for FDR. Is it Benjamini-Hochberg method?*

The FDR procedure used was indeed Benjamini-Hochberg. However, in our new analyses we now switched to Bonferroni correction, to ensure more robust and interpretable results (see our reply to the next comment). FDR-correction is now only used in two places in the exploratory gene-set interaction analysis, we have clarified in the Methods and main text that in those instances Benjamini-Hochberg was used.

3. *The ultimate goal can be to allow users to use this pipeline in their studies. To do this, a careful step-by-step description of the pipeline (or manual) should be available. Moreover, it will be great if as many parts of the pipeline are automated. For example, you can automate the generation of QQ plot and give outputs of how much each QQ is deviated? (e.g. like genomic-control deviation factor).*

We have made a number of changes to achieve this. Firstly, the workflow itself was clarified and refined, with the initial GSA and exploratory interaction analysis now included as part of the workflow as steps 1 and 6. We also changed from using FDR correction to Bonferroni correction as, unlike FDR, this does not depend on which variables are being tested, therefore making it much easier to compare across analysis steps or maintain the same threshold in different steps. It also has the subsetting property, which FDR correction lacks.

More explicit recommendations have also been added on various thresholds and cut-offs. We have updated step 4 (set-specific QQ-plots), adding a 95% confidence band to the plots and defining an additional metric based on it, to aid interpretation. Step 6 (exploratory interaction) has also been more clearly defined as part of the analysis workflow, also adding testing for negative interaction to better complement the rest of the workflow (as this can help detect associations that are innately difficult to detect in the initial GSA in step 1).

In addition, we have moved some parts of the Methods section describing the analysis steps into the Supplemental Information and have rewritten it into an extensive analysis guideline. This elaborates both on how to perform the analysis and the decisions that need to be made in the process, as well as going into how results can be interpreted. This guideline is written largely in terms of recommendations rather than a strict protocol because the analysis still needs to be guided by researcher interpretation. However, it should provide the researcher with sufficient structure to perform the analysis. This guideline is in addition to the detailed step-by-step description of the analysis of the blood pressure phenotypes we performed for this study, which should provide additional guidance. The Methods section also contains an overview of the analysis workflow, and how each step is performed.

Automation for some of the components of the analysis workflow are available in MAGMA 1.07 (this is not yet released, but will be prior to publication of this paper), though the need for researcher input makes it difficult to fully automate. The output necessary for creating set-specific QQ-plots and scatterplots for continuous variable by set interactions can be requested from the

program, in a format that can easily be used in programs like R. An R script with functions for reading in this output and creating the requisite plots will be released alongside MAGMA 1.07.

4. *Sup Figure 2B is missing?*

This has been fixed.

REVIEWER 2

The authors previously developed a tool MAGMA for testing gene set enrichment for genomic studies including genome-wide association studies. In current manuscript, they extended the framework to conditional analysis and interaction analysis. The primary application of the tool is to better understand which gene set is truly associated with the trait when multiple gene sets are overlapping. While it is potentially useful for GWAS secondary analysis, I have several comments.

1. One of the claim of the manuscript is that the conditional approach has the potential to discovery “causal” enrichment. This is misleading. The causality based on mediation analysis typically requires explicit biological hypotheses. For example, SNP A is associated with lung cancer risk and smoking; smoking is associated with lung cancer risk. A mediation analysis can be done using conditional analysis to establish the causality of the SNP A. For current manuscript, the conditional analysis is including two gene sets with overlapping genes. I do not see the way to make any claim about causality. While the conditional analysis helps reduces the number of associations, the author need to think of another way to interpret the results and the benefits of this analysis. Need to clearly define “causality” and “spurious findings” in this setting.

We agree that ‘causal’ is too strongly phrased, as a real causal conclusion cannot be drawn from these analyses. We intended to refer to a genuine involvement of gene properties in (the genetic etiology of) a phenotype. We have removed the terms ‘causal’ and ‘spurious’ from the paper and supplementals (except for unrelated uses of ‘causal’). Where necessary, it has been replaced by more neutral phrasings like ‘plays a role in the phenotype’ or ‘involved in (the genetic etiology of) the phenotype’.

2. Type-I error need to be checked. The authors have applied stepwise analysis (1) standard GSA for marginal association (2) for significant associations perform conditional and interaction analysis. Simulations are required to access the type I error rate for the whole procedure.

We have added a simulation study to evaluate the components of the analysis workflow. As the statistical performance of the gene analysis and the core gene-set analysis framework have previously been evaluated (De Leeuw et al. (2015) and De Leeuw et al. (2016), these are references 5 and 1 in the manuscript), we focused specifically on the conditional GSA and interaction GSA models. We did not include a separate evaluation of the stepwise procedure, as this is used only to discard previously selected gene properties (based on significance in step 1) rather than select new ones. The simulations are described in the sections *Results - Simulations*, *Methods - Simulation Study* and *Supplementary Note - Simulation study*.

3. As a methodology paper, large scale simulations are important to understand the statistical

property of the test, which includes but not limited to (A) the way of summary gene-level Z-score based on SNP PCA analysis (is this approach powerful?), (B) when gene set A is selected if A and B are overlapping and are both associated (enriched) with the trait and (C) interaction testing.

As per the previous comment, we have included a simulation study to evaluate the performance of the conditional and interaction GSA models under a range of different scenarios. We evaluated both the type 1 error rate of the models as well as how the models (including the marginal model) behaved, and biases that can result, if the model was misspecified relative to the simulation scenario. The focus of the simulations was on different scenarios at the gene-set level, as the GSA models are defined to be agnostic with regard to the gene analysis model that computes the gene Z-score and the differences between those models have been previously studied (De Leeuw et al. 2015, reference 5 in the manuscript).

4. Application to blood pressure reduced the number of significant enrichment. How do we know the new results are scientifically more plausible? For these significant results, replication study is preferred in an independent GWAS data set.

In part the plausibility derives from the structure of the model, as conditional analyses (both using linear regression and more generally) are a known and well-studied way of disentangling the relations between multiple correlated variables. They have so far not really been applied to the GSA context, but the same statistical principles apply.

Of course, being certain of the plausibility of the results ultimately requires functional analyses confirming the suggested biological hypotheses. But short of that a replication in independent GWAS data (and possibly converging evidence from other types of data such as differential expression) are indeed an invaluable, and usually more attainable, resource.

As suggested, we included a replication of our results (Supplementary Note - Replication analysis). Unfortunately, only one previous GWAS study that we could find (ICBP, 2011) published full SNP summary statistics (some others did publish subsets of top associated SNPs, but the GSA requires SNP statistics that are not filtered on their p-values). Because these results are relatively old they are based on a much smaller sample size and contain far fewer SNPs (and therefore also somewhat fewer genes). Consequently the correlation in gene Z-scores between the ICPB and UKB data was low, around 0.3 for SBP and DBP and about 0.21 for PP.

Although ICBP was thus not ideal as a replication sample, on aggregate the replication can nevertheless be considered largely successful. Of the 67 separate associations that were analysed in the replication analysis, 91% had a p-value below 0.5 and 48% was nominally significant, far more than would be expected by chance. And indeed several of the associations were also successfully replicated individually, including some of the interactions (all replication results can be found in Supplementary Tables 6, 7 and 8).

In addition to the replication, there is now also a section in the Supplemental Note on GSA results from previous GWAS papers (section *Gene-set analyses in recent blood pressure GWAS*). Those results are largely quite difficult to compare due to differences in the annotation used in particular. However, the Liu et al. (2016) study also analysed Gene Ontology gene sets, with no known overlap with in data with the UK Biobank sample. The results of their GSA indeed shows considerable similarity with ours, which helps to further support the plausibility of our results.

5. While MAGMA made it clear to be applicable to GWAS summary data, the authors did not mention whether the current approach can be used to GWAS summary data. If this analysis cannot be done based on GWAS summary data, the impact will be limited.

The GSA framework in MAGMA is independent of the details of how the input gene analysis is performed, so can be used with any gene analysis model and any input data that the gene analysis can accept, including analysis of GWAS summary statistics. This is now also explicitly mentioned in the Methods section, at the end of the *Conditional and interaction gene-set analysis model* subsection.

Minor comment:

You have two “Figure 1”.

This has been fixed.

REVIEWER 3

The authors provide a new extended approach to gene-set analysis, by incorporating conditional and interaction analyses, in contrast to the traditional GSA approach which only assesses the marginal associations of gene properties. This enables the identification of confounding as well as the discovery of novel gene properties that would not be detectable marginally, altogether improving the robustness, reliability and specificity of the results and conclusions. And the results can be restricted only to those which are more likely to be causal. This is good work, well done.

This approach is applied to blood pressure phenotypes from the UK Biobank data, as an example, considering tissue expression data from GTEx, miRNA target data and Gene Ontology data. They illustrate that many of the significant results from traditional GSA would lead to misleading conclusions for gene properties which were not actually likely to be causally relevant, or that associations were not tissue-specific. The refined conclusions for blood pressure were interesting, for example with the ability to show distinct differences between the three different blood pressure phenotypes, which are not usually revealed through BP GWAS papers.

Major comments:

1) Whilst I agree with the authors that “the analyses of blood pressure phenotypes provide novel insights into their genetic etiology”, I think the paper is lacking an initial literature review of what has been reported so far in this context from blood pressure genetic analyses. I was surprised that none of the recent BP GWAS papers were cited within the references. In fact, the reference list seems somewhat short. Many recent BP GWAS papers have performed pathway analyses, expression and enrichment analyses as secondary analyses to their GWAS analyses. It would therefore be helpful, for the readers and research community, if the authors could briefly compare the GSA approach to other such approaches, e.g. IPA or DEPICT, etc, from a methodological perspective. Then, also, contrast the final conclusions here on significant gene properties, with those that have been previously reported in recent BP genetics papers. The authors here do compare their refined results to what would have been obtained from traditional GSA alone, but only within the context of their own analysis of the UKB data. It would also be interesting to compare to what was already present in the literature.

We have added an overview of recent GWAS literature at the end of the introduction, and have also added an overview of GSA results from those papers and a comparison of those results with our own in the Supplementary Note (section *Gene-set analyses in recent blood pressure GWAS*). The comparison is somewhat complicated by the difference in data, analysis methods and in

particular in the input annotation used in those analyses however, though a clear similarity in results is found with the Liu et al. (2016) paper.

To our knowledge no other GSA methods are able to perform conditional or interaction analysis, including IPA and DEPICT. As a result there is in that regard nothing we can directly compare our approach to.

2) *In the Online Methods we realise that this analysis was only performed using the 2015 interim N~150k release of the UKB data. As many submitted journal papers are now using the full July 2017 N~500k UKB data, I think that it needs to be made clearer at the start of the paper, that this only uses the interim data. Ideally, time permitting, could all analyses be updated now to the full N~500k UKB data? Or at least, please acknowledge this in the discussion, and perhaps comment on how the authors expect the accuracy of results or final conclusions may or may not change with the increased sample size and newly imputed genetic data?*

We have now rerun our analyses with the full UK Biobank (July 2017) data, replacing all results from our previous analyses.

3) *Please provide a more thorough detailed description of your own QC of the UKB data. The opening paragraph of the Supplementary text is still very brief. For example:*

- a. *What method / criteria did you use to remove non-European individuals?*
- b. *What degree of relatedness did you use for exclusions of related individuals?*
- c. *Please state the exact number of samples excluded according to each individual exclusion criteria separately? E.g. N for non-Europeans, relatives, sex-discordant, etc?*
- d. *I note that the INFO threshold of 0.8 is much stricter than the imputation quality cut-offs usually used in GWAS. Please comment and explain why a stricter threshold was used for GSA here?*

The data and QC section has been updated to contain this information.

The reason for using a much stricter INFO threshold for gene and gene-set analysis is that due to the aggregation of SNPs to genes and then sets, it is not possible to determine post-analysis if and to what extent a result might be influenced by poorly imputed SNPs. It is therefore important to filter those out in advance instead. We have now added this motivation explicitly in the Methods section, in *Genotype and phenotype data* (note also that for the new UK Biobank data, we used an INFO threshold of 0.9 rather than 0.8).

4) *From the Supplementary text, I note that data was converted to best-guess genotype format PLINK data for analysis. Please comment and explain why imputed dosage data cannot be used within these GSA analyses? And if not, please discuss this as a possible limitation?*

The main reason for this is that (at present) MAGMA can only use binary PLINK format data as input, and not BGEN-format dosage data. Analysis of dosage data can still be used indirectly however, since MAGMA can also analyse SNP summary statistics. SNP analysis of the dosage data can therefore be performed in the requisite software, with the resulting SNP p-values then being fed into MAGMA. Reference data would still be provided in PLINK format, but as this is only used to estimate LD the loss of information that would incur would be quite limited. The only restriction in this case would be that the PC regression gene analysis model cannot be used, but for analysing high coverage imputed data the SNP-wise (multi) model is recommended in any case.

The possibility of running the analyses based on summary statistics from any kind of single SNP analysis is now explicitly noted in *Methods - Conditional and interaction gene-set analysis model*.

5) *The three most highly implicated tissues for BP remaining seem to be the heart, coronary artery and uterus. Please could you provide further discussion on the biological interpretation of these final findings? For example, whilst the first two may be less surprising for blood pressure as a cardiovascular phenotype, perhaps you could offer an interpretation of the relevance of the uterus tissue expression?*

We have added an extensive discussion on the biological implications of our results to the Discussion section, as well as some additional subsections in the Results to better support that discussion.

6) *The authors used MAGMA software for the initial GSA. Have these conditional and interaction extensions of the GSA framework to the software also been made publicly available as software tools, to enable other analysts to use this proposed analytical approach?*

All the components needed for the analysis (except plotting) are implemented as part of MAGMA 1.07. This MAGMA update will be released prior to publication, alongside an R script that can be used for the QQ-plots and scatterplots used as checks in the analysis workflow (input data for those plots is provided by MAGMA in plain-text format, so other software can be used for plotting as well). We have also now added an analysis guideline in the supplementals, which should further aid researchers in performing these analyses on their own data.

Minor comments:

1) *On pg.5 you say that additional analysis and follow-up research would still be required for the other four sets. Please could you discuss more details of what these additional analyses would be?*

With the reanalysis for the changed data this paragraph is no longer there, but the same general point is now discussed in more detail in *Analysis guideline - Step 4* (and specific to our new results in the Detailed overview of blood pressure analyses) in the Supplementary Note. In general this will vary; sometimes the interaction analyses in step 5 may already reveal an interaction that explains signs of confounding in the QQ-plot, in other cases it may be necessary to analyse additional, more detailed annotation related to the gene set and the genes it contains to see if a confounder can be detected.

2) *I note an error in the Figure 1 legend, where left and right are the wrong way around.*

This has been fixed.

3) *In Figure 2, please state the significance thresholds used (i.e. in addition to stating within the text).*

Figure 2 is only meant as a general workflow and is not specific to our blood pressure analysis, hence why no specific significance thresholds are given. The analysis workflow is in principle compatible with any standard multiple testing correction that can be applied to the initial GSA in step 1, so this is up to the individual researcher. Recommendations for the thresholds to use are

provided in the Analysis guideline in the Supplementary Note. The thresholds that were used in our own blood pressure analyses are given the Methods section.

4) Pg.14 shows a third Figure, which is labelled as "Figure 1", so I am assuming this should be titled Figure 3?

It should, the Figure labels have now been fixed.

5) On pg.20 the authors helpfully give warnings regarding the analysis of "small gene sets" which are "particularly vulnerable". Please could you give more detail as to how small is "small"?

We now suggest an absolute minimum of 50 genes, though recommend at least a 100 genes. Note that this section has been moved to the supplementals, integrated in the extensive analysis guideline we added (in *Analysis guideline - interaction analysis* in the Supplementary Note).

6) Please state the MAF variant filter in the Online Methods too, not only in the Supplementary file, so that the reader can see where the ~21M variants have come from. Furthermore, as the MAF threshold was 0.1%, please could you discuss any implications of using rare variants within these GSA analyses?

The MAF threshold is now listed in the Methods section as well, in the *Genotype and phenotype data* subsection; with the change to the full UK Biobank data, we have also lowered the threshold to 0.00001.

Rare variants are dealt with using a built-in burden score mechanism. This aggregates rare variants (specified by a user-defined MAF or MAC threshold; we used an MAC of 100) into one or more burden scores per gene, and replaces the rare variants with these burden scores. This is described in the *Primary gene-set analysis* subsection of the Methods. MAGMA also has options to filter out rare variants internally, and it should be noted that the SNP-wise (mean) model is largely equivalent to the inverse-variance weighted version of SKAT. How rare variants will end up affecting any particular analysis will depend on the genetic architecture of the phenotype, so it is difficult to say in general what the implications would be. The gene-set analysis framework is agnostic in this regard however, so it is not dependent on any particular way of handling rare variants in the gene analysis.

7) It then becomes confusing that even though ~21M variants are stated to be in the dataset (pg. 21), that then only ~9M variants can actually be annotated to genes (pg. 22). Does this mean that the GSA analyses then only actually consider these final ~9M variants? Please clarify.

The analysis is indeed only directly dependent on a subset of the variants, as only variants that are mapped to a gene are actually used. The mapping is based on the transcription region of the gene plus an optional window around that, so any variants located too far from any gene are not used at all. In practice this is usually around 40 - 45% of the variants in the input depending on window size (it is 43.7% in our current analyses). This is now more explicitly noted in the *Annotation* subsection of the Methods.

8) Please justify the covariates that you have used? (pg. 21) I note that BMI is not adjusted for, even though this covariate is usually used within most BP GWAS analyses?

The choice of covariates was based on which covariates are typically used in BP GWAS, as well as the typical covariates needed to correct for stratification artefacts. The omission of BMI was an oversight, but this has now been included in our new analyses.

9) For any readers who are less familiar with MAGMA, please also state all the criteria used, even though all the settings were at “default values” (pg. 23)

Of the default values referred to, in this case only a second parameter of the burden scoring mechanism pertains to our analyses. There are also various internal QC settings that are intended as safeguards against issues missed in the regular SNP QC, but these are rarely changed. To remove potential confusion we now mention the second burden scoring parameter (maximum number of variants per score) explicitly (in the *Primary gene-set analysis* subsection of the Methods) so all settings that inform the analysis are listed, and removed the reference to default values.

Reviewer #1 (Remarks to the Author):

The authors addressed the issues, and I have no further comments.

Reviewer #3 (Remarks to the Author):

The authors have made substantial improvements to this revised manuscript, well done.

I particularly commend the authors on the following improvements:

- using the new version of UK Biobank data
- good additions to the Introduction and Discussion sections
- new simulation study
- new replication study (although this does not seem to feature many comments in the main manuscript, only the Sup Note, see comments below)
- extra sections to the Sup Note, e.g. the new Analysis Guidelines section, and the "Gene-set analyses in recent blood pressure GWAS" section
- extra details on the QC of the data

This is now a great detailed Methods paper, which is well written, and will be a good contribution to the field.

I only have the following minor comments:

Main manuscript:

- 1) In lines 76-78, the simulation study and UK Biobank analysis are mentioned. So please also mention here in the Introduction, that a replication study was also performed.

2) Likewise, I can't clearly see any mention in the Methods section to the Replication study that was performed. So it seems to only appear in the Sup Note. Unless I missed it, please include a brief comment in the Methods section on this, and then cite the Sup Note.

3) The sub-headings of the Results section are not clear to follow as well directed signposting. The first sub-heading clearly states "Simulations". But then, after this, there is no clear sub-heading to show when the Results move from the simulation study to the main UK Biobank BP analysis.

Sup Info:

4) Despite the contents page, there seem to be no page numbers in the actual Sup Info doc, so please add these

5) One typo on pg.25 in the "Comparison" section, where there is a missing closing bracket in the sentence about the three cardiovascular system results.

REVIEWERS' COMMENTS:

Reviewer #1 (Remarks to the Author):

The authors addressed the issues, and I have no further comments.

Reviewer #3 (Remarks to the Author):

The authors have made substantial improvements to this revised manuscript, well done.

I particularly commend the authors on the following improvements:

- using the new version of UK Biobank data*
- good additions to the Introduction and Discussion sections*
- new simulation study*
- new replication study (although this does not seem to feature many comments in the main manuscript, only the Sup Note, see comments below)*
- extra sections to the Sup Note, e.g. the new Analysis Guidelines section, and the "Gene-set analyses in recent blood pressure GWAS" section*
- extra details on the QC of the data*

This is now a great detailed Methods paper, which is well written, and will be a good contribution to the field.

I only have the following minor comments:

Main manuscript:

1) In lines 76-78, the simulation study and UK Biobank analysis are mentioned. So please also mention here in the Introduction, that a replication study was also performed.

This has been added.

2) Likewise, I can't clearly see any mention in the Methods section to the Replication study that was performed. So it seems to only appear in the Sup Note. Unless I missed it, please include a brief comment in the Methods section on this, and then cite the Sup Note.

This has been added to the *Genotype and phenotype data* subsection of the Methods, referring to the Supplementary Information. A reference to the ICBP data has been included in the reference list, and the subsequent references 40 and 41 were renumbered to 41 and 42.

3) The sub-headings of the Results section are not clear to follow as well directed signposting. The first sub-heading clearly states "Simulations". But then, after this, there is no clear sub-heading to show when the Results move from the simulation study to the main UK Biobank BP analysis.

We have renamed the header of the *Simulations* subsection to be more in line with the style of the other subheaders (ie. a brief summary of the main point of the subsection), to emphasize that only this particular section pertains to the simulation results.

Sup Info:

4) Despite the contents page, there seem to be no page numbers in the actual Sup Info doc, so please add these

Page numbering has been added to the Supplementary Information.

5) One typo on pg.25 in the "Comparison" section, where there is a missing closing bracket in the sentence about the three cardiovascular system results.

This has been fixed.